# In Situ Treatment of Refractory Perianal Fistulas in Dogs with Low-Dose Allogeneic Adipose-Derived Mesenchymal Stem Cells

**DOI:** 10.3390/ani14223300

**Published:** 2024-11-16

**Authors:** Nathaly Enciso, Javier Enciso-Benavides, Juan Sandoval, Javier Enciso

**Affiliations:** 1Grupo Medicina Regenerativa, Universidad Científica del Sur, Lima 150142, Peru; 2Clínica Veterinaria Enciso, Lima 15039, Peru; javier.enciso.benavides@gmail.com; 3School of Veterinary Medicine, Faculty of Health Sciences, Universidad Peruana de Ciencias Aplicadas, Lima 15023, Peru; 4Facultad de Medicina Veterinaria, Universidad San Luis Gonzaga de Ica, Ica 11770, Peru

**Keywords:** mesenchymal stem cells, cell therapy, canine, perianal fistula

## Abstract

Adult stem cell therapy has demonstrated anti-inflammatory and immunomodulatory effects in many diseases in dogs. This quality gives them great potential for application in immune-mediated diseases such as perianal fistula, where conventional treatments do not guarantee cure. Although there are publications on the use of stem cells in the treatment of this disease, they are scarce and there is no consensus on the application protocol, nor has there been medium and long-term follow-ups, so it is useful to propose new protocols. In this paper, we report a protocol using low doses of adipose stem cells from healthy donors, applied directly to the lesions, which was effective after 1–3 months in all the patients treated (100%), in most cases with a single application and with a follow-up duration of effect of 12–48 months. This proposed protocol is clinically simple, effective, long-lasting, safe, and inexpensive, which is in line with the expectations of dog owners.

## 1. Introduction

Perianal fistulas (PFs) are painful ulcers that appear spontaneously on the skin surrounding the anus of dogs of any breed, with German Shepherds being the most affected (more than 80% of the population), followed by beagles at 4% [1], while in other breeds, no incidences have been described. They are an immune-mediated condition in which the current standard treatment is pharmacological. Cyclosporine is the recommended treatment of choice, but it may be associated with numerous adverse [2] and surgical effects. This condition can be debilitating for the affected dogs and can lead to euthanasia if not treated effectively [3]. Canine perianal fistula (PF) is considered a condition analogous to human perianal Crohn’s disease (CD), as the anatomy and size in dogs (especially large breeds) is very similar to that of humans, allowing the dog to be used as an animal model to evaluate new treatments and subsequently conduct human clinical trials [4].

Since PFs are immune mediated, they have several characteristics in common with the fistulae in human Crohn’s disease (CD) [5]. Considering the recent advances in the medical and surgical treatment of CD, remission rates remain low and, despite the development of new biological therapies, the average efficacy at one year is 30–50% [6]. The factors involved in the pathogenesis of CD fistulising disease include an increase in the production of transforming growth factor β and interleukin-13 in the inflammatory infiltrate that induce epithelial to mesenchymal transition and an increase in matrix metalloproteinases, leading to tissue remodelling and fistula formation [7,8].

However, it has been shown that adult mesenchymal stem cells (MSCs) mediate immune suppression through various mechanisms, such as immune modulation and the suppression of inflammation [9]. Many studies have evaluated the efficacy of these stem cells in immune-mediated diseases in humans and animals [10].

The efficacy and safety of expanded allogeneic MSC therapy for CD in humans has been demonstrated in phase III clinical trials [11], and a patent has been obtained in Europe [12], suggesting that the treatment of spontaneous canine PF with pluripotent stem cells could be used as a model to evaluate cell therapy [3]. However, after revision of the PubMed database, only one open trial carried out using embryonic MSC in canines with good results was found [4,12]. Nevertheless, the data available support the hypothesis that the results from testing cell therapies in dogs with anal furunculosis have a significant translational value in optimising MSC transplant procedures in patients with perianal CD (pCD) [4].

Given the few publications on the use of allogeneic adipose stem cells for the treatment of PF in canines, and the lack of consensus protocols, the aim of this pilot clinical trial was to evaluate the efficacy and safety of a new protocol proposed by our working group related to the administration of expanded and cryopreserved allogeneic adipose stem cell therapy for spontaneous PFs in dogs.

## 2. Materials and Methods

### 2.1. Experimental Animals

This trial was conducted from January 2019 to December 2023 at the Enciso Veterinary Clinic, which was performed according to the Animal Research: Reporting In Vivo Experiments (ARRIVE) guidelines, was approved by the Ethics Committee of the San Luis Gonzaga National University of Ica, Peru (Resolution No. 247-CE-FMVZ-UNICA-2018), and informed consent was obtained from the owners for the treatment.

The evaluation was carried out with fourteen dogs of different breeds between 4 and 16 years of age; 64% (9/14) were males and 36% (5/14) were female and all had a clinical diagnosis of perianal fistula (PF); six had simple fistulas and eight had complex fistulas based on Sandborn’s description [13] (Table 1). All animals were previously treated with conventional drug therapy using cyclosporin A (2–10 mg/kg) for a period of 12 months, with all the cases presenting early relapse. To participate in the study, cyclosporine treatment was stopped for one month prior to the use of ASCs. Exclusion criteria were the presence of significant clinical conditions/diseases and previous surgery for the treatment of perianal fistula.

The patients underwent a rigorous clinical examination to rule out the presence of abscesses and tumours. If tumours were diagnosed, the patients were excluded. If an abscess was detected, antibiotic therapy with a combination of ciprofloxacin (10 mg/kg/12 h) and metronidazole (25 mg/kg/12 h) was administered and, after remission of the abscess, the patient was then included in the trial.

### 2.2. Histological Evaluation

To corroborate the diagnosis of a perianal fistula by comparing the histological lesions with the previous studies, a biopsy of the ulcers was taken using lidocaine chlorhydrate (VETOCAINE^®^. Agrovet Market, Lima, Peru) as a local anaesthetic, which was infiltrated into the ulcer orifice using a 1 mL syringe with a 26 × ½′ needle. Immediately after taking the tissue sample of the ulcer with a 4 mm diameter biopsy punch, it was placed in a flask with 10% formalin solution in phosphate buffer. Histological processing was performed according to the routine protocol, which, in summary, consisted of embedding the tissue in paraffin to obtain 5 µm thick histological sections and staining with haematoxylin and eosin [14]. The histopathological study was then performed under an optical microscope (Nikon Eclipse E200, Tokyo, Japan).

### 2.3. Adipose Stem Cells

Adipose-derived mesenchymal stem cells (ASCs) were obtained from healthy donor female dogs undergoing elective sterilisation and processed at the Good Manufacturing Practice (GMP) facility of the Universidad Científica del Sur. The cells were preserved in liquid nitrogen for 12 months in the cryobank of the Stem Cell Laboratory of the Enciso Veterinary Clinic in Lima (Peru). The isolation, culture, characterisation, differentiation, and cryopreservation of ASCs were performed as described by Enciso et al. [15]. Briefly, the ASCs were cultured in DMEM with 10% foetal bovine serum (FBS) and 1% antibiotics for expansion. At the fifth passage, when they reached adequate confluence (>80%), they were maintained in medium without FBS for 48 h. Aliquots were taken for cell counting, viability assays, microbiological tests, and the characterisation and differentiation of ASCs into the three lineages (osteogenic, chondrogenic, and adipogenic tissue), which were confirmed by the following stains: Von Kossa staining, Alcian blue staining and Oil Red O staining (respectively); the samples were finally frozen at −80 °C.

### 2.4. Quality Assurance of ASCs

On the day of cell administration to the patient, the cells were thawed according to a previously described routine procedure [16] with phosphate-buffered saline (PBS), pH 7.4, used as a diluent for administration. ASCs used for injection were cultured in the Luria–Bertani medium (Scharlab S.L., Barcelona, Spain) to evaluate sterility. Before using the ASCs therapeutically, the cells were analysed by flow cytometry to confirm that they were positive for CD90 and negative for CD34, CD45, and MHC-II, and the expression of *GAPDH*, *TBP*, *OCT4*, *RUNX2*, *SOX9*, and *PPARγ* genes was detected by Reverse-Transcription PCR (RT-PCR) as previously reported [17].

Quality control of the cells included quantitative polymerase chain reaction (qPCR) tests for *Mycoplasma*, *Ehrlichia canis*, and *Anaplasma* sp.; protocols used were described by Barker et al. 2010 and Gaunt et al. 2010 [18,19] with negative results. These tests were performed prior to freezing and before use as a treatment. Cell viability was evaluated using trypan blue (with a viability greater than 95%), and the solution was distributed into 1 mL syringes with 25 G × 5/8″ needles at a concentration of 1 × 10^6^ cells per syringe. The syringes were kept at 4–8 °C until cell administration.

### 2.5. Allogeneic ASCs Administration Procedure

Prior to the ASC administration procedure, the general anaesthesia of the patient was induced by intravenous neuroleptoanalgesia using a combination of xylazine/ketamine (1 mg/kg/10 mg/kg, Ket-A-Xyl^®^, Agrovet Market, Animal Health, Lima, Peru). Subsequently, the syringes with the cells were removed from refrigeration and kept at room temperature in the operating room of the veterinary clinic, until administration in doses of 1 × 10^6^ diluted in PBS at each point of each fistula, as indicated in Figure 1, using a 1 mL syringe with a No. 25 G × 5/8″ needle. Since the origin of a fistula is the internal orifice, it was considered that efficacy would be greater if the cells were injected into the tissue surrounding the fistula orifice and along the walls of the fistulous tracts; care was taken not to deposit the cells in the lumen of the orifice, similar to previous work in humans [20]. In no case was the fistula sutured and neither topical nor systemic antibiotics were applied. A second and third dose was applied in two patients (Table 1. Patient number 4 and 10) when there was evidence of recurrence of the clinical picture expressed by pruritus in the perianal area, but no fistula was present. A dose of 5 × 10^6^ cells was applied to the erythematous area.

### 2.6. Clinical Evaluation of the Effect of ASCs

The efficacy endpoint was established 3 months after injection and follow-up was performed in some cases up to week 48. Efficacy was defined as a complete closure of the fistulous tract and internal and external openings without drainage or signs of inflammation. The presence or absence of recognised local and systemic adverse clinical events such as fever, urticaria, pruritus, and possible tumour development in the vicinity of the fistulae was assessed.

On days 0 (ASCs injection), 7, 30, 60, and 90, patients were clinically examined to determine the presence/absence and depth of fistulae. In addition, to determine the safety of the treatment, a haemogram and blood biochemistry were performed to assess liver and renal function.

### 2.7. Statistical Analysis

The results were analysed using SPSS 25 (IBM Corporation, Endicott, NY, USA) and Graph Pad Prism version 6 (Software, Inc., San Diego, CA, USA). Data distribution was analysed using the Shapiro—Wilk test. Comparison between the number of fistulas pre-treatment and at 1, 3, 6, and 12 months after treatment with allogeneic ASCs was evaluated with ANOVA and Tukey’s post hoc test for multiple comparisons. All data are expressed as the mean and standard deviation. A difference was considered significant when the *p* value was <0.05.

## 3. Results

The thawed ASCs used for this trial had the minimum characteristics to be considered mesenchymal stem cells, according to the International Society for Tissue Cell Therapy (ISCT) [21]. The cells presented a fibroblastoid morphology and adhered to plastic (Figure 2(A.1)) and differentiated into osteocytes, chondrocytes, and adipocytes (Figure 2(A.2), (A.3) and (A.4) respectively). In addition, the cells were phenotyped as CD90 positive while less than 2% were CD34, CD45, and MHCII positive (Figure 2B) and showed the expression of stem cell markers *OCT4*, *RUNX2*, *SOX9*, and *PPARγ* (Figure 2C).

Eleven patients (79%) received a single dose of the ASCs, two patients (14%) received two doses, and one patient (7%) received three doses over a period of 90 days due to the severity of the lesions, although the fistulas closed within one month, signs of inflammation remained; and after the last dose, no recurrence was shown until 12 and 24 months in the patients who received two doses and 48 months in the patient who received three doses (Figure 3A). The difference in post-cell therapy follow-up is due to the different presentation times of the cases within the study period.

In the study period, out of a total of fourteen patients, 100% had no recurrence at 12 months post-treatment. By the end of the study of the fourteen patients, five (36%) patients had no recurrence 13–24 months after treatment, while the fistula did not recur in four patients between 25 and 36 months after treatment (29%), and three patients (21%) had no recurrence between 37 and 48 months after treatment (Figure 3B). A majority (64%, n = 9) of the patients were male (Figure 3C).

The therapeutic protocol described in the present study achieved a therapeutic efficacy of 100% in the treated dogs, presenting a complete resolution of the fistula lesions within 3 months, and with remission after one month in some cases (Figure 4). There was a significant difference in the number of fistulas before treatment compared with the number of fistulas at 1, 3, 6, and 12 months after treatment with allogeneic ASCs (*p* < 0.0001) (Figure 5).

Histological study showed a predominance of active chronic inflammation with epidermal ulcers. In two cases, the lesions were deeper, presenting pyogranulomatous cellulitis with the involvement of the adnexal glands. The inflammatory infiltrate was composed of neutrophils, lymphocytes, macrophages, and plasma cells, and the presence of necrotic areas (Figure 6). However, there was no uniformity regarding the presence of acute and chronic inflammatory cells. For humanitarian reasons, no post-therapy biopsies were performed by request of the owners of the patients.

During the months of follow-up of the study patients, no local or systemic adverse side effects, such as fever, hypersensitivity, altered blood cell counts or liver and kidney function, and no tumour development were observed.

## 4. Discussion

The results of this work demonstrate that cryopreserved allogeneic ASCs are effective for the treatment of refractory canine PFs, with no recurrence or adverse effects observed in patients for varying periods of time ranging from 12 to 48 months post-treatment in 100% of the cases studied. Our trial is the first to be conducted with this type of cell at low doses (5 × 10^6^), but higher than those which were suggested by Kim et al. [22] (1 × 10^6^ cell/cm^2^), as well as repeated doses in only two out of the fourteen cases at 3 months post cell therapy, with a prolonged follow-up to monitor for recurrence or adverse effects. In contrast to a previous study in dogs that used doses of 2 × 10^7^ human embryonic cells/fistula [10], which is higher than in our work, the dogs took 3 months to become fistula-free and the study had one case of recurrence at 6 months, which was the follow-up time in this work. In another experimental study in rabbits with induced perianal fistulas, they used a rabbit adipose stem cell line at a dose of 2 × 10^6^ cells/fistula, but the fistulas were sutured beforehand, healing occurred 6 days after therapy and they had a post-therapy follow-up of only 70 days [23].

It is also important to note that the dose used in this study is five times lower than the dose applied in situ in a recent study involving humans with the same cell type [24]. Although we have no explanation for this remarkable dose difference, perhaps the dose effect is later but more prolonged in humans compared to canines. However, this cannot be verified, as the follow-up time after cell therapy in both studies was medium term. Nevertheless, a recent review study found that the difference in cell types, cell sources, and cell dose did not influence the mid-term efficacy of MSCs in humans [25], but we believe that the difference in dose may be explained by having different interspecies immune response mechanisms.

Regarding the efficacy of ASC administration in the PFs reported in the present study, it could be explained by the same mechanisms described for immune-mediated and inflammatory diseases, since the histological study of the present cases showed the involvement of chronic active inflammation coincident with the descriptions of Killingsworth et al. [26], as previously proposed by Day [17,27]. This could be explained by the fact that MSCs have been shown to be effective in the treatment of multiple immune-mediated diseases in humans and in animals with spontaneous disease [9]. Likewise, the mechanism of action of stem cells has been demonstrated in inflammatory and immune-mediated processes [8] in refractory wounds in dogs [17] and in canine atopic dermatitis [15,28]; all of these diseases are caused by inflammatory and immune-mediated processes, similar to the pathogenesis of canine perianal fistula.

Moreover, despite the small sample size of the present study, we found that 100% of cases responded satisfactorily, with 79% (11/14) of the ASC-treated dogs requiring only one dose, and 86% (12/14) were relapse-free for at least 24 months after treatment. One patient has been relapse-free for 48 months. When compared to the current approach, we see that for many years, PFs have been approached with surgical procedures that, aside from not being curative, can lead to the development of collateral damage such as anal stenosis and faecal incontinence [29,30], while in pharmacological treatment, which is currently the treatment of choice, medical protocols use cyclosporin A but cannot guarantee the cure of the disease [13]. In fact, not all cases respond satisfactorily and pharmacological treatment can be associated with numerous adverse effects [2]. Moreover, less satisfactory long-term results can be observed, such as the recurrence of lesions in 30–50% of dogs treated with cyclosporin A [31].

In this study, however, during the 12 to 48 months of follow-up, no adverse effects were observed in any of the cases studied, so we can affirm that the ASCs used in this study are innocuous and did not induce any adverse disorder. This quality of the cells used in this study could be explained by the absence of immunogenicity due to the lack of MHC class II and very low levels of MHC-I [32,33]. This biological quality could contribute to the standardisation of application protocols and the dose used, as suggested in other studies [34]. Also, by having a long-term follow-up of up to 4 years, it decreases the risk of the main concern regarding the administration of MSCs, which is the possibility that they may promote tumour development by inducing neoplastic cell proliferation and neoangiogenesis [35], although no cases of tumour development have been described yet [36]. On the other hand, based on these results and as in humans, we recommend that, in the future, the effect be evaluated over longer periods to see if there are relapses of the perianal fistulas.

Although this study included a small number of patients (n = 14), it is the largest evaluation of the use of low-dose allogeneic adipose stem cells in cases of spontaneous perianal fistula and with a medium- and long-term follow-up, considering the life expectancy of the canine species. In addition, it is pertinent to note that due to the economic limitations of the owners, MRIs could not be routinely performed during the follow-up of the patients. However, our results may have translational value for optimising MSC transplantation procedures in human patients with perianal Crohn’s disease, taking into account the low dose of stem cells used in this work compared to the high doses used in humans described in other studies [4,37].

## 5. Conclusions

In conclusion, the results of this study show that the exclusive administration of allogeneic cryopreserved adipose stem cells at a low dose of 5 × 10^6^/fistula according to the new protocol described in this study completely resolved perianal fistulas 1–3 months after in situ application in 100% of the cases studied. There was no recurrence of the initial clinical manifestations for 12–48 months, depending on follow-up time, and no local or systemic adverse effects were observed.

## Figures and Tables

**Figure 1 animals-14-03300-f001:**
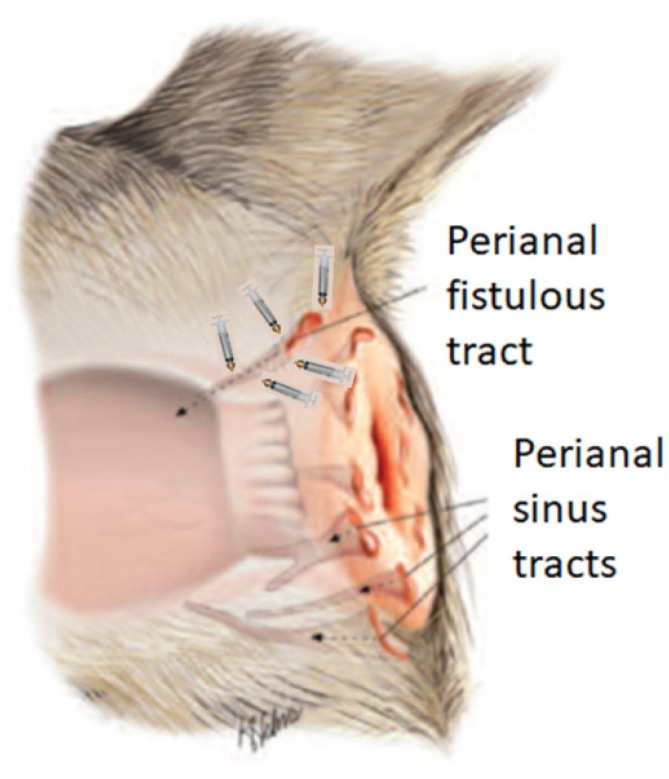
Diagram of local administration of adipose stem cells around the fistula. The syringes indicate the fistula sites where the adipose stem cells were applied. Adapted from the Illustration by Mr. Kerry Helms in Patterson, A.P., Cambell, K.L. (2005) with permission of the author [13].

**Figure 2 animals-14-03300-f002:**
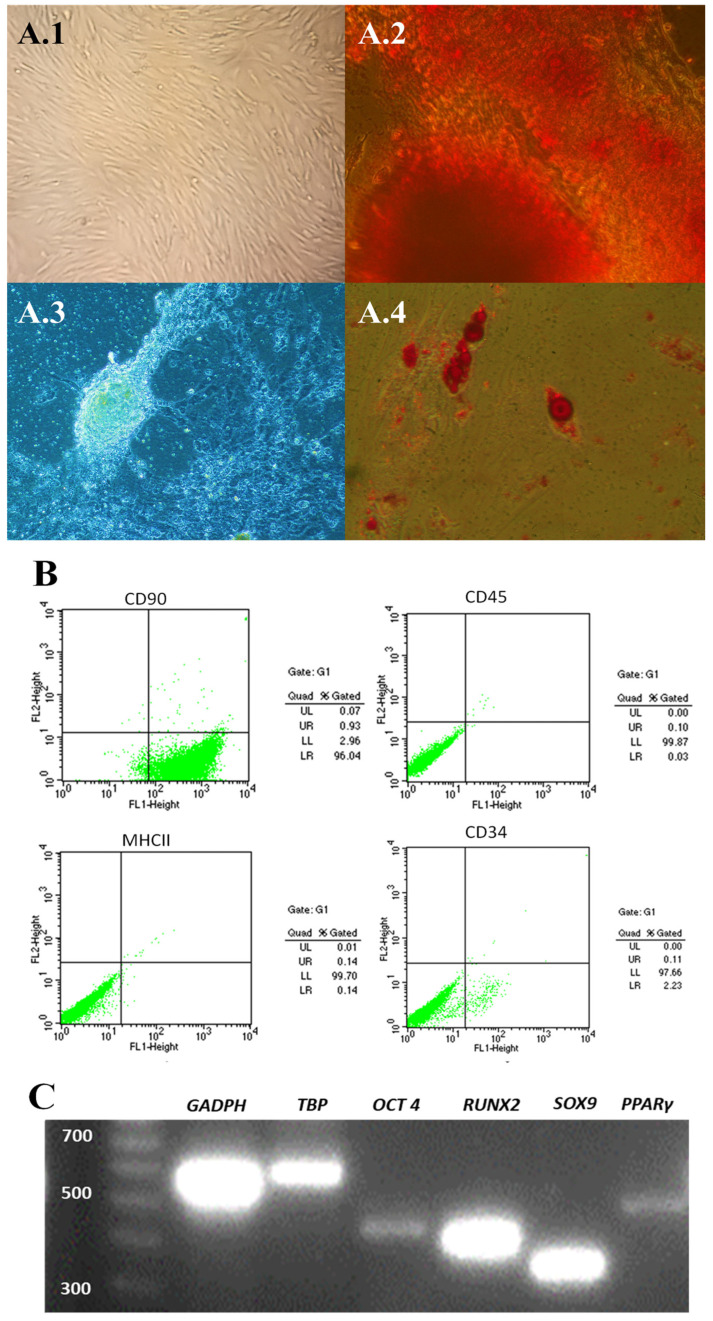
Characterisation and differentiation of canine adipose-derived mesenchymal stem cells (ASCs). (**A**) Differentiation of ASCs (**A.1**) into osteogenic (**A.2**), chondrogenic (**A.3**) and adipogenic (**A.4**) lineages. Cells were stained with von Kossa stain for osteogenic lineage (2), Alcian blue stain for chondrogenic lineage (3) and oil red O stain for adipogenic lineage (4). (**B**). Phenotypic characterisation of ASCs by flow cytometry. Results positive for CD90 and negative for CD34, CD45 and MHCII. (**C**). Expression of pluripotency markers in ASCs. The expression genes were *OCT-4*, *RUNX2*, *SOX9,* and *PPARγ*, while *GAPDH* and *TBP* were used as housekeeping genes.

**Figure 3 animals-14-03300-f003:**
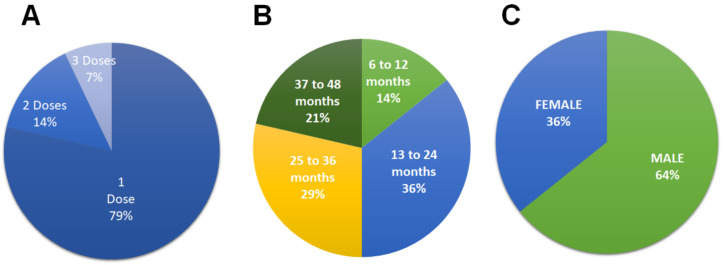
Representation of the study population with regard to the number of doses (**A**); months after treatment (**B**); and sex (**C**). n = 14.

**Figure 4 animals-14-03300-f004:**
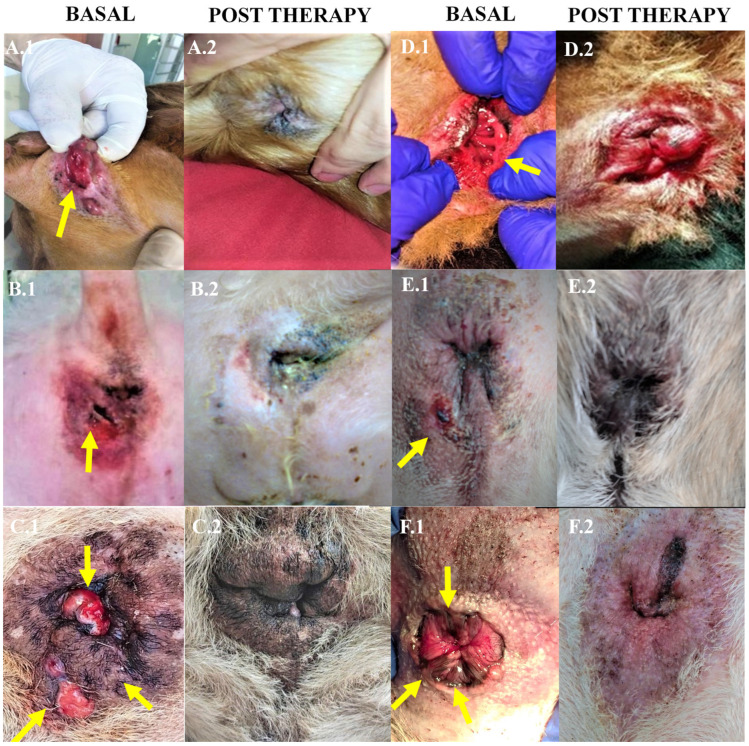
Representative images showing lesions from the basal stage (**A.1**–**F.1**) and three months’ post therapy (**A.2**–**F.2**) to the total healing of six clinical cases of perianal fistula treated with cryopreserved allogeneic adipose stem cells. Yellow arrows indicate perianal fistulas.

**Figure 5 animals-14-03300-f005:**
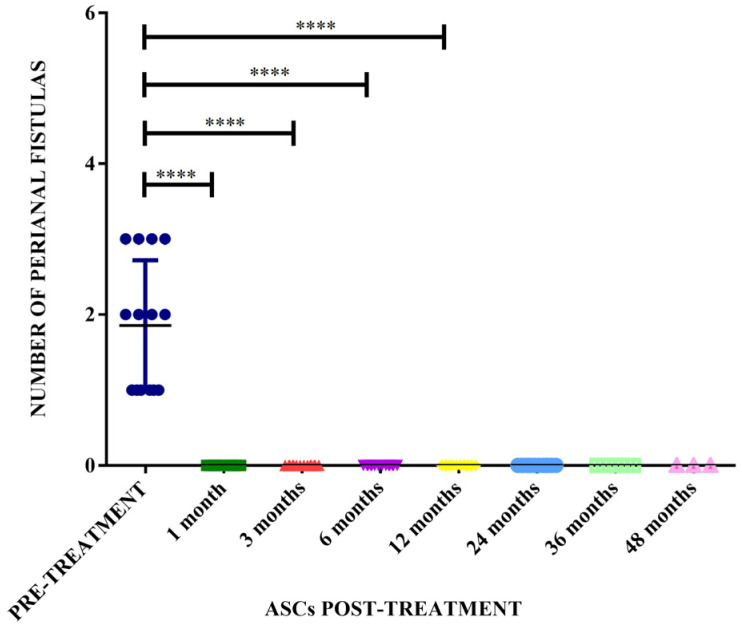
Comparison between the number of fistulas presented pre-treatment and the number of fistulas after treatment with allogeneic adipose stem cells (ASCs) and statistical analysis between pre-treatment and 1, 3, 6, 12, 24, 36, and 48 months post-treatment. **** means a significant difference of *p* < 0.0001.

**Figure 6 animals-14-03300-f006:**
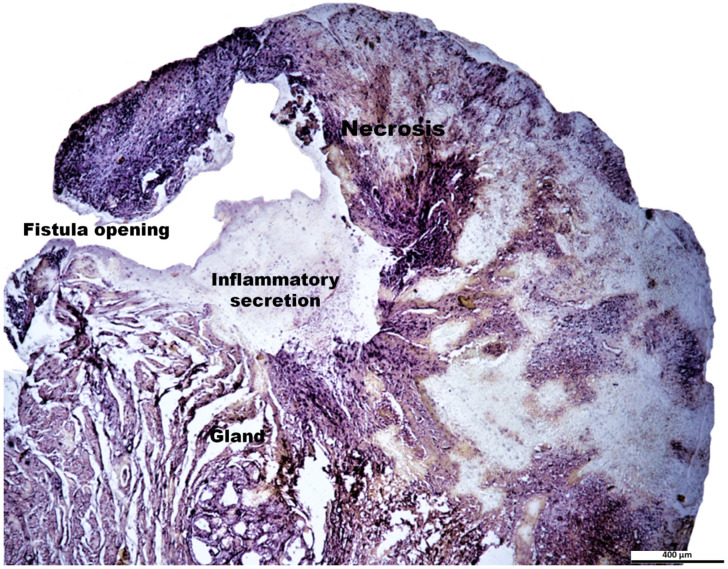
Histopathological representation of the biopsy of the perianal fistula before ASCs treatment. The microphotograph shows the opening of the fistula, accumulation of inflammatory secretions in the saccule and adjacent tissues, as well as necrosis and chronic inflammation in the epidermis. Bar scale = 400 µm. Staining with haematoxylin and eosin.

**Table 1 animals-14-03300-t001:** Clinical characteristics of the dogs participating in the trial, dose per fistula and cell numbers.

Case (#)	Age (Years)	Breed	Sex	# Fistulas	# Cells Applied/Dose	# Doses	Follow-Up (Months)
1	15	Jack Russell	Male	3	15 × 10^6^	1	12
2	4	Boston Terrier	Female	1	5 × 10^6^	1	12
3	13	Mongrel	Female	2	10 × 10^6^	1	18
4	14	Pug	Male	3	15 × 10^6^	3	24
5	10	Golden Retriever	Male	3	15 × 106	1	24
6	16	Jack Russell	Male	2	10 × 10^6^	1	24
7	11	Jack Russell	Male	1	5 × 10^6^	1	24
8	12	Bichon Frisé	Female	1	5 × 10^6^	1	30
9	12	Jack Russell	Male	1	5 × 10^6^	1	30
10	10	Pug	Male	3	15 × 10^6^	2	30
11	6	Pug	Male	1	10 × 10^6^	1	36
12	7	Springer spaniel	Female	2	10 × 10^6^	1	44
13	7	Maltese	Female	1	10 × 10^6^	1	45
14	4	Cocker spaniel	Male	2	10 × 10^6^	2	48

## Data Availability

Data are contained within the article.

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
