# Peer review of "In Situ Treatment of Refractory Perianal Fistulas in Dogs with Low-Dose Allogeneic Adipose-Derived Mesenchymal Stem Cells"

_animals, 2024, doi:10.3390/ani14223300_

Round 1
Reviewer 1 Report (Previous Reviewer 2)
Comments and Suggestions for Authors
The authors have replied my concerns one by one. The quality of the manuscript is now greatly improved. The reviewer has no more comments.
Author Response
We are grateful for your time in reviewing this manuscript and value your recommendations for improving the manuscript.
Reviewer 2 Report (Previous Reviewer 3)
Comments and Suggestions for Authors
Attached in the file

Round 2
Reviewer 2 Report (Previous Reviewer 3)
Comments and Suggestions for Authors
NA
This manuscript is a resubmission of an earlier submission. The following is a list of the peer review reports and author responses from that submission.
Round 1
Reviewer 1 Report
Comments and Suggestions for Authors
1. The study is conducted in dogs but a general term of canine is used, which represents a larger group of animals. Therefore, the focus should only be on dogs, and "Canine" should be replaced with "Dogs".
2. Section 2.1 the heading "patient" needs to be replaced with experimental animals.
3. Figure 2, chondrocytes show no staining, it means cells were not differentiated. Additionally, flow cytometer data is shown for only one positive marker (CD90) and three negative markers, while the authors are required to show three positive and one negative marker. Similarly, the gene expression data need to be shown through real time PCR.
4. Figure 3 is not required at all, author need to mention these details in methods.
5. Methodology for Figure 6 is not provided, no justification that how histology is performed, or how the sample is obtained from a live animal.
6. The major limitation of the study is the missing of control, over three months the normal healing system of the body will work and heal these fistulae. The author didn't compare these results with control (not treated) dogs.
7. The study miss scientific justification and pathway that the proposed therapy work, no scientific explanation is provided.
Comments on the Quality of English LanguageMinor editing
Reviewer 2 Report
Comments and Suggestions for Authors
As the authors stated, perianal fistulas (PFs) are spontaneously occured in dogs. The effective treatments of PFs is important. Since other methods may have side effects, the authors evaluated the effects of using low-dose allogeneic adipose tissue-derived mesenchymal stem cells, although based on a relatively small patient population (14). Overall the manuscript is well organized and written in fluent English. A few minor issues need to be checked:
Line 47-48: Do perianal fistulas (PFs) only occer in dogs? What's the incidence of PFs in dogs?
Line 109-112: "After the fifth passage, aliquots were taken for cell counting, viability assay, microbiological tests, characterisation and differentiation of the ASCs into the three lineages (osteogenic, chondrogenic and adipogenic tissue) and finally frozen at -80 °C." This description is too simple. The authors need to provide more detailed information for three lineage staining (although it's breifly mentioned in legend of Fig 2)or a reliable reference.
Line 125,132, 141, 235: check "1x106" "5x106" "2x107".
Line 154-156: "In addition, to determine the safety of the treatment, a haemogram and blood biochemistry were performed to assess liver and renal function." Where is the results of haemogram and blood biochemistry?
Line 156: "In addition, clinical evaluation of patients who started the trial continued". please check the syntax, it's difficult to understand.
Comments on the Quality of English Language
The quality of English is good. But a few sentences are difficult to be understood.
Reviewer 3 Report
Comments and Suggestions for Authors
The manuscript titled “In situ treatment of refractory canine perianal fistulas with 2 low-dose allogeneic adipose tissue-derived mesenchymal stem cells” is well written and presents some novel information. However, some major concerns need to be addressed before further processing.
Minor Concerns
1. The manuscript contains numerous instances of typographical errors and abbreviations, as exemplified in the following lines:
- In line 118, the abbreviation "MCHII" is used instead of "MHCII," as noted in the FACS figure.
- Similarly, in line 172, the correct symbol for PPARγ should be utilized.
- Additionally, in the discussion section, the term "PF" should be replaced with "FP."
- It is also observed that exponential digits are not properly superscripted.
- It is imperative to meticulously proofread the manuscript before submission to ensure the elimination of these errors.
Major Concerns
2. Should PF be classified as an immune-mediated or autoimmune disease? Considering that the author has successfully managed this disease with stem cell therapy, how likely is it that PF will recur given the commonness of relapses in autoimmune diseases?
3. How did the authors determine that no immunogenic response occurred after the injection of allogeneic stem cells?
4. What evidence supports the efficacy of the stem cells in terms of cell viability after injection into the perianal fistula? How can the authors confidently assert that the closure of fistulas and the recovery observed post-treatment is because of the cells initially implanted into the perianal fistula?
5. It is unexpected that no immune reaction was observed following the administration of allogenic stem cells or the continuation of cyclosporine treatment by the authors, which is contrary to the standard protocol for in situ or therapeutic stem cell studies. Typically, in such studies, autologous cells are employed to circumvent potential immune reactions in the patient. Furthermore, FBS (Fetal Bovine Serum) is not utilized in cell culture, instead, serum-free media is preferred. It is imperative to explain the rationale behind deviating from these established practices.
6. The method employed by the authors to determine the dosage of stem cells for injection into the PF typically involves a calculation based on the length and diameter of the affected area. This approach ensures that the dosage is tailored to the specific dimensions of the fistula, optimizing the efficacy of the treatment.
7. In the patient section of the Materials and Methods, the authors have documented a total of 14 dogs, all of whom are male. This information is noted on line 87. However, upon reviewing Table 1, it appears that there are also female dogs included in the dataset.
